# Efficient recognition of facial expressions does not require motor simulation

**Gilles Vannuscorps[1,2,3]\*, Michael Andres[2,3], Alfonso Caramazza[1,4]**

[1]Department of Psychology, Harvard University, Cambridge, United States; [2]Institute of Neuroscience, Université catholique de Louvain, Ottignies-Louvain-la-Neuve, Belgium; [3]Psychological Sciences Research Institute, Université catholique de Louvain, Ottignies-Louvain-la-Neuve, Belgium; [4]Center for Mind/Brain Sciences, Università degli Studi di Trento, Mattarello, Italy

**Abstract** What mechanisms underlie facial expression recognition? A popular hypothesis holds that efficient facial expression recognition cannot be achieved by visual analysis alone but additionally requires a mechanism of motor simulation — an unconscious, covert imitation of the observed facial postures and movements. Here, we first discuss why this hypothesis does not necessarily follow from extant empirical evidence. Next, we report experimental evidence against the central premise of this view: we demonstrate that individuals can achieve normotypical efficient facial expression recognition despite a congenital absence of relevant facial motor representations and, therefore, unaided by motor simulation. This underscores the need to reconsider the role of motor simulation in facial expression recognition.

## Introduction

Stereotyped facial movements – facial expressions – reveal people's happiness, surprise, fear, anger, disgust, and sadness in a way that can be easily interpreted by their congeners (*Ekman, 1982*). The ability to express and recognize facial expressions is crucial for social interactions (*Elfenbein et al., 2007*). A fundamental issue addressed here concerns the nature of the mechanisms underlying facial expression recognition: what types of information are used in recognizing a facial expression in everyday life?

On one view, the efficient recognition of facial expressions relies on computations occurring in the visuo-perceptual system, supported by perceptual processes and information extrapolated from perceptual learning (*Bruce and Young, 1986*; *Du et al., 2016*; *Huelle et al., 2014*). Three types of representations are necessary and sufficient for efficient visual recognition of facial expression: (1) a series of progressively more abstract visuo-perceptual representations of the postures and movements of the observed face; (2) stored structural descriptions of the features characterizing known facial expressions; and (3) the conceptual/semantic representations that characterize the facial expressions (e.g., *Bruce and Young, 1986*). Of course, viewing facial expressions may bring about other cognitive, affective and motor processes, involving episodic memories, empathy or imitation, but these other types of processes and representations are not necessary for efficient recognition of facial expressions.

In the last 20 years, an alternative view suggesting that efficient (i.e., fast and accurate) facial expression recognition cannot be achieved by visual analysis alone but requires a process of motor simulation – an unconscious, covert imitation of the observed facial postures or movements – has gained considerable prominence (*Goldman and Sripada, 2005*; *Ipser and Cook, 2016*; *Montgomery and Haxby, 2008*; *Niedenthal et al., 2010*; *Pitcher et al., 2008*; *Paracampo et al., 2017*). This 'motor' view has become increasingly influential in neuroscience, philosophy, neurology and psychiatry where it is suggested that it opens new clinical perspectives for the diagnosis,

**\*For correspondence:**
gilles.vannuscorps@uclouvain.be

**Competing interests:** The authors declare that no competing interests exist.

understanding and rehabilitation of clinical populations presenting with facial expression recognition deficits such as Parkinson's disease (*Ricciardi et al., 2017*), autism (*Dapretto et al., 2006*; *Gordon et al., 2014*), and schizophrenia (*Torregrossa et al., 2019*).

Three main types of evidence are typically cited in support of the motor theories. First, neuroimaging studies and studies combining transcranial magnetic stimulation (TMS) of the motor cortex and electromyographic recording of facial muscles have reported that viewing a facial expression activates parts of the motor system, which are also involved in executing that facial expression (*Carr et al., 2003*; *Dimberg, 1982*; *Dimberg et al., 2000*; *Hess and Blairy, 2001*; *Leslie et al., 2004*; *Montgomery and Haxby, 2008*). Second, performance in tasks involving facial expression recognition can be modulated by experimental manipulations of the normal state of the observers' motor system, such as concurrent motor tasks or transcranial magnetic stimulation (TMS) over the motor cortex (*Ipser and Cook, 2016*; *Maringer et al., 2011*; *Oberman et al., 2007*; *Paracampo et al., 2017*; *Ponari et al., 2012*; *Rychlowska et al., 2014*; *Wood et al., 2016*). Third, studies of individuals suffering from congenital or acquired facial expression production disorders have been reported to have difficulties in recognizing facial expressions. Several studies have reported, for instance, that individuals with Moebius Syndrome, an extremely rare congenital non-progressive condition (*Verzijl et al., 2003*) resulting in facial paralysis (usually complete and bilateral), scored below average in facial expression recognition experiments (*Bate et al., 2013*; *Calder et al., 2000*; *Giannini et al., 1984*; *Nicolini et al., 2019*). Co-occurrence of facial expression production and recognition disorders have also been reported in Parkinson's (*Ricciardi et al., 2017*) and in Huntington's disease (*Trinkler et al., 2013*).

However, these various findings are open to alternative explanations. The activation of the motor system during the observation of facial expressions could result from a mere reaction (e.g., emotional contagion or a way to signal empathy) of the observer to others' facial expression *after* it has been recognized as an instance of a given emotion (*Hess et al., 2014*). It may also result from visuo-motor transformations serving other purposes than facial expression recognition such as working memory encoding (*Vannuscorps and Caramazza, 2016a*) or mimicry in order to foster affiliative goals (*Fischer and Hess, 2017*).

The demonstration that interfering with the observer's motor system may influence facial expression recognition performance in some tasks clearly shows that the motor and visual systems are functionally connected. However, the conclusion that these findings demonstrate that the motor system is necessary for efficient facial expression recognition faces two main problems. First, the results do not imply that the motor and the visual systems are functionally connected for the purpose of facial expression recognition. TMS applied to an area may have distant effects on other areas to which it projects (*Papeo et al., 2015*; *Ruff et al., 2009*; *Siebner et al., 2009*; *Siebner and Rothwell, 2003*). The motor cortex projects to the visual cortex involved in the perception of biological motion in order to support the control of one's body movements (*Wolpert et al., 2003*) and the visual cortex involved in the perception of body parts shows increased activity when moving these body parts even in the absence of visual feedback (*Astafiev et al., 2004*; *Dinstein et al., 2007*; *Orlov et al., 2010*). Thus, TMS applied to the motor system may have functional effects upon facial expression recognition by modulating the visuo-perceptual system to which it is naturally connected to support the control of one's movements. Second, and more importantly, evidence that information in one system (e.g., the motor system) may influence computations in another system (e.g., the visual system) is not evidence that the former is necessary for the latter to function efficiently. For instance, the finding that visual information about lip movements influences auditory speech perception does not imply that auditory speech perception requires the visual system and has never been presented as a proof of this (*McGurk and MacDonald, 1976*). Likewise, results showing that auditory information affects the efficiency of facial expression recognition have been interpreted as evidence that available auditory or context information can influence facial expression recognition, but not that the auditory system is critical for efficient facial expression recognition (*Collignon et al., 2008*). Thus, although the effects of TMS and behavioral motor interference provide interesting information about perceptual and motor interactions, they may very likely originate from outside the set of cognitive and neural mechanisms that are necessary for efficient facial expression recognition. Thus, the behavioral, fMRI and TMS results reported above are not critical to discriminate between the perceptual and motor view because both views can equally account for the finding that interfering with the motor system can modulate the efficiency with which facial expressions are recognized.

Equally indeterminate is the evidential value of the reported co-occurrence of deficits of facial expression production and recognition. Such co-occurrence indicates that a relationship exists between production and perception abilities, but on their own such instances do not imply that there is a causal relationship between them. The interpretation of these associations is all the more difficult given that most patients with Parkinson's and Huntington's disease have widespread brain lesions and suffer from cognitive disorders like visuo-perceptual or executive function disorders (*Ricciardi et al., 2017*; *Trinkler et al., 2013*), which may be independently responsible for poor performance in facial expression recognition. In line with this possibility, in a study of 108 patients with focal brain lesions, *Adolphs et al., 2000* found no association between the ability to recognize facial expressions and motor lesions/impairment. Likewise, Moebius syndrome typically impacts not only the individuals' sensorimotor system, but also their visual, perceptual, cognitive, and social abilities (*Bate et al., 2013*; *Carta et al., 2011*; *Gillberg and Steffenburg, 1989*; *Johansson et al., 2007*). Therefore, it is not clear whether the facial expression recognition deficit observed in these cases was the direct consequence of the production disorder or, instead, the result of other impaired but functionally separate processes. In a study by *Bate et al., 2013*, for instance, five out of six participants with Moebius Syndrome were impaired in at least one of three facial expression recognition tasks. It may be tempting to conclude that this finding supports motor theories. However, the same five individuals were also impaired in their ability to recognize facial identity and/or in tests assessing low-level vision and object recognition indicating that the facial expression recognition impairment cannot unambiguously be associated with the facial paralysis.

In favor of the view that recognition does not require motor simulation, there are reports of individuals with Moebius Syndrome (IMS) who achieve a normal level of performance in facial expression recognition despite their congenital facial paralysis (*Bate et al., 2013*; *Calder et al., 2000*; *Rives Bogart and Matsumoto, 2010*). For instance, in contrast to the five other MS participants from Bate et al.'s study (2103) cited above, one IMS (the only one who obtained normal scores in facial identity recognition, object recognition and low-level vision tests, MB4) performed very close to the average level of performance of the controls in three facial expression recognition tests (the Ekman 60 faces test, the Emotion Hexagon test, and the Reading the Mind in the Eyes test) despite having facial movements restricted to slight puckering of the mouth bilaterally. In *Rives Bogart and Matsumoto, 2010* study, the performance of 37 IMS participants was comparable to that of 37 control participants in a task in which they viewed pictures of one of seven emotions (anger, contempt, disgust, fear, happiness, sadness, and surprise) and were asked to select the corresponding emotion from a list of the alternatives.

Although these findings undermine the hypothesis that facial expression recognition requires motor simulation, it has been suggested that a mild or subtle deficit in facial expression recognition may have gone undetected in these studies because they relied mostly on untimed picture labelling tasks (*De Stefani et al., 2019*; *Van Rysewyk, 2011*). In addition, these previous findings left open the possibility that motor simulation may contribute to facial expression recognition efficiency when the tasks are more challenging, such as when facial expressions are more complex, must be interpreted quickly, only partial information is available, or when the task requires subtle intra-category discriminations such as discriminating fake versus genuine smiles (*Niedenthal et al., 2010*; *Paracampo et al., 2017*). The research reported here was designed to explore this large remaining hypothesis space for a role of motor simulation in facial expression recognition: is it *possible* to achieve efficient facial expression recognition without motor simulation in the type of sensitive and challenging tasks that have been cited by proponents of the motor theories as examples of tasks in which motor simulation is needed to support facial expression recognition?

We studied 11 individuals with Moebius Syndrome using an experimental procedure designed to overcome the sensitivity and interpretative issues that have been raised for previous reports of intact facial expression recognition abilities in IMS:

1. Given the heterogeneity of the clinical expression of Moebius Syndrome, especially in terms of associated visuo-perceptual symptoms (*Bate et al., 2013*; *Carta et al., 2011*), and the specific prediction of the motor simulation hypothesis tested in this study – viz., that none of the individuals with congenital facial paralysis should be as efficient as the controls in facial recognition – we conducted analyses focused on the performance of each IMS.
2. To ensure the sensitivity of the facial expression recognition measures and the specificity of the assessment with respect to the predictions of the motor theories, we selected five

challenging facial expression tasks that have been shown to be sensitive to even subtle facial expression recognition difficulties and/or that have been explicitly cited by proponents of the motor theories as examples of tasks in which motor simulation should support facial expression recognition (see Materials and Procedures for detail).

3. To minimize the likelihood of false negatives, that is, the risk to conclude erroneously that an IMS achieves a 'normotypical level' of efficiency in a facial expression recognition experiment, we compared the performance of each IMS to that of 25 typically developed highly educated young adults and selected a high alpha level (p>0.2), which set the threshold for 'typically efficient' performance in a given experiment to scores above 0.85 standard deviations below the mean of the controls *after* control participants with an abnormally low score were dismissed (i. e., a score below two standard deviations from the mean of the controls).

4. To gain more power to detect even subtle degradations of facial expression recognition skills than in simple IMS versus controls comparison, we also examined each participant's performance in two experiments assessing facial identity recognition abilities and in one experiment assessing the ability to recognize emotions from emotional speech. This allowed us to assess the impact of the inability to engage in motor simulation on facial expression recognition skills in a within-subject design, by comparing the IMS's ability to recognize facial expressions versus facial identity and emotional speech recognition, in comparison to the controls' performance for these tasks.

5. To investigate the possibility that the IMS who achieve typically efficient performance in facial expression recognition could do so by completing the task through different means or somewhat differently than the control participants, we included planned qualitative analyses of the performance of the IMS who achieved quantitatively normotypical performance profiles.

Using the outlined experimental procedures, we established three criteria for concluding that an individual achieves 'normotypically efficient facial expression recognition': (1) s/he scores above 0.85 standard deviations below the mean of the controls in *all* the facial expression recognition tasks, (2) s/he does not perform significantly worse in these tasks than in other face-related visual tasks (face identity) and emotion recognition tasks (vocal emotion recognition), (3) and, s/he performs these tasks in a way that is qualitatively similar to the controls. If motor simulation were necessary for efficient facial expression recognition, then, none of the tested IMS should meet these criteria.

## Results

The results of all participants are displayed in *Figure 1*. We conducted six series of analyses. We first conducted analyses to verify the sensitivity of the facial expression recognition tasks and the representativity of our sample of control participants. To this end, we first performed a series of one-sample *t*-tests to verify that the controls' performance was significantly above chance and a series of Shapiro-Wilk tests to verify that the controls' data were distributed normally in all the facial expression recognition experiments. The results of these analyses showed that the controls' data were indeed significantly above chance (all *t*-tests > 8) and normally distributed in all five facial expression experiments (all five Ws > 0.930, all ps > 0.13). Then, we compared the results of our control participants to those of comparable samples available in the literature. The aim of these analyses was to verify that our control sample was representative. Previous data were available for Experiments 3–5. One-sample *t*-tests indicated that our control participants' average performance (M = 28.6; SD = 2.4) in Experiment four was significantly better than that reported in the only other study reporting the results of the same (French) version of the task (M = 24.8; SD = 3.8; *t* (24)=7.9; *Prevost et al., 2014*). Our control group also performed slightly better (M = 63.04, SD = 10.22) than the average performance of the three comparable age groups reported in a previous study using the task of Experiment 3 (M = 62.6, SD = 9.6; *t* (24)<0.2; *Kessels et al., 2014*). Our control sample performed lower than the performance expected from *Paracampo et al., 2017* norming study of the Amusement from Smiles experiment (66% in our study vs 75%). However, our experiment included only half the number of trials of the original experiment (*Paracampo et al., 2017*) offering less perceptual learning opportunity to our participants (*Huelle et al., 2014*). Taken together, the outcomes of these analyses thus provided confidence in the sensitivity of the tasks and in the representativity of the controls' data.

In a second step, we turned to our main question and investigated whether at least one IMS performed at a 'normotypical' level of performance (at least above 0.85 standard deviation below the

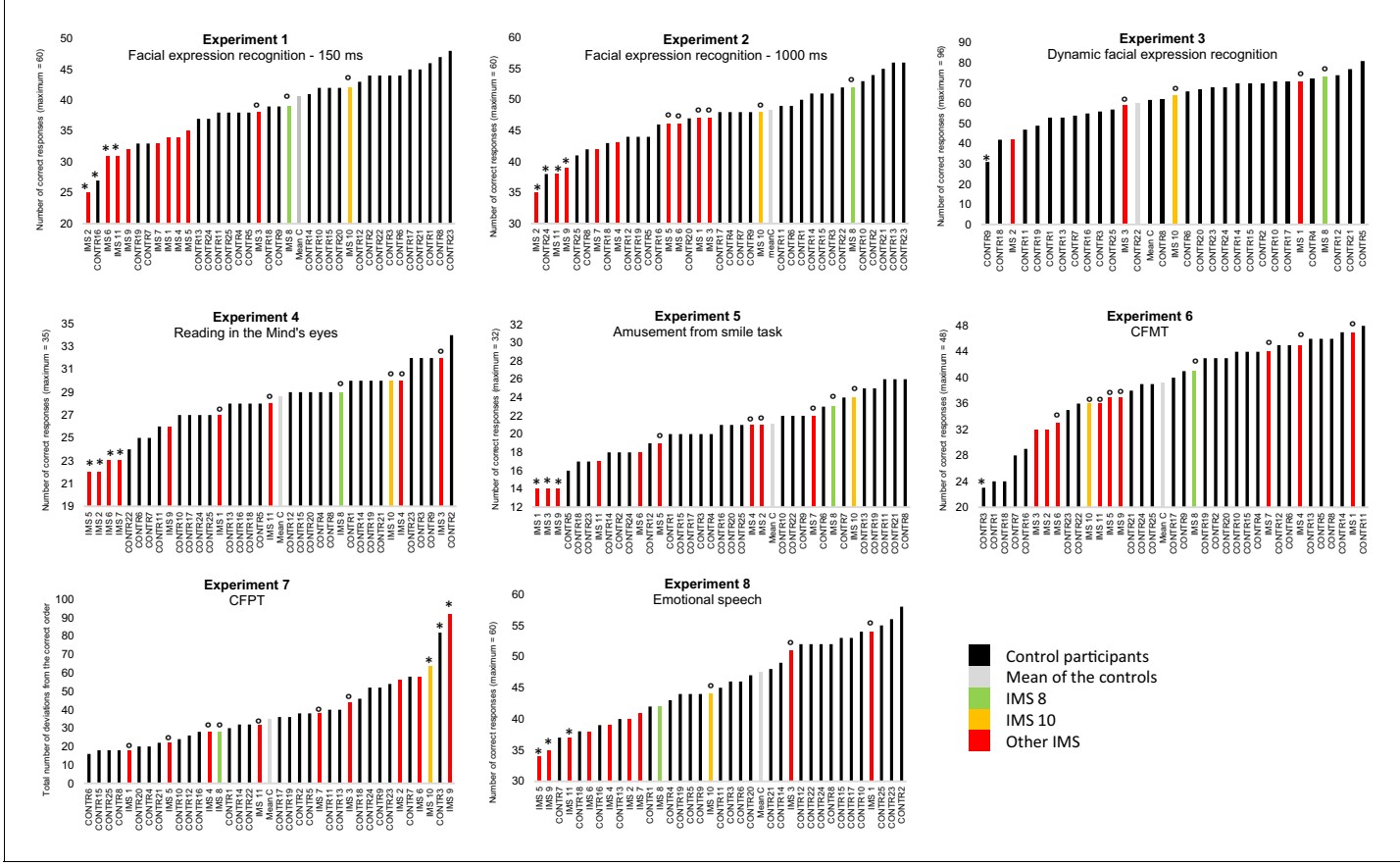

**Figure 1.** Results of Experiments 1–8 by individual participant. In black: control participants; in light grey: mean of the controls; in green and yellow: IMS 8 and 10 who performed Experiments 1–5 with normotypical efficiency; in red: the nine other IMS. A small circle (°) indicates IMS participants with a 'normotypical' score (at least above 0.85 standard deviation below the controls' mean performance) after control participants with an abnormally low score (below 2 SD from the other control participants) was/were discarded (indicated by an *). An asterisk (*) also indicates IMS participants with a score below two standard deviations from the mean of the controls.

controls' mean performance) in the five facial expression experiments. To obtain a stringent test of performance equality, we first excluded from the control sample participants who performed below two standard deviations from the controls' mean (one participant in Experiment one and one participant in Experiment 2, indicated by an * in *Figure 1*). Then, a series of *Crawford and Howell, 1998* modified t-tests were used to test whether each IMS's performance in the various facial expression recognition experiments was or was not 'normotypical' (indicated by an ° in *Figure 1*). These analyses indicated that IMS 8 and 10, displayed in green and yellow in *Figure 1*, performed the five experiments with normotypical efficiency.

Third, we focused on the two IMS who met the criteria for normotypical performance and assessed whether any discrepancy between their performance and that of the controls was larger on facial expression recognition (Experiments 1–5) than on facial identity (Experiments 6–7) and speech emotion recognition (Experiment 8). The aim of these analyses was to seek evidence that despite their normotypical performance on facial recognition tasks, IMS 8 and 10 may nevertheless be comparatively less good in facial expression recognition than in other tasks not assumed to rely on motor simulation. To this end, we computed Crawford and Garthwaite's Bayesian Standardized Difference Test (BSDT) (*Crawford and Garthwaite, 2007*). The BSDT allows computing an estimate of the percentage of the control population exhibiting a more extreme discrepancy between two tasks than a given individual. We performed 15 BSDTs for each IMS (comparison of three control tasks and five facial expression recognition tasks of interest). All the comparisons were either clearly not significant (3/30 comparisons, all BSDTs > 0.6) or indicated a comparatively better performance in facial expression recognition tasks than in the control tasks (27/30 comparisons). Thus, there was no evidence

that IMS 8 and 10 performed facial expression recognition tasks less efficiently than facial identity recognition or emotional speech recognition tasks.

Fourth, we conducted a qualitative analysis of the performance of these two IMS (IMS 8, 10) in the facial expression recognition experiments. The aim of these analyses was to seek evidence that despite their quantitative normotypical performance, these IMS might have performed the task somewhat differently than the controls; for instance, that they used different facial diagnostic features. Any such processing differences would likely result in different patterns of behavioral responses to expressions and, in particular, in different patterns or errors. To explore this possibility, we computed response matrices between the six displayed facial expressions (in rows) and the six response alternatives (happiness, surprise, anger, sadness, fear, and disgust) for Experiments 1–3 separately for the control participants and for IMS 8 and 10 (*Figure 2*). Then, to examine the similarity between the controls' and the IMS's matrices, we vectorized the matrices and correlated them with each other. The resulting correlation coefficients were very high both when all responses were considered (all three Pearson's Rs (36)>0.95; all *p*s < 0.001) and when only errors were considered (all three Pearson's Rs (30)>0.63; all *p*s < 0.001), indicating that the groups consistently confused the same set of alternatives. Note that the confusion matrices of the other IMS were also strongly correlated to that of the control participants (all three Pearson's Rs (36)>0.95; all *p*s < 0.001; see *Figure 2—figure supplement 1*).

Fifth, we conducted additional tests and analyses of IMS 8 and 10's facial movement abilities to explore the possibility that, despite their severe facial paralysis, they could nevertheless imitate some aspects of the facial expressions tested in Experiments 1–5, which would provide them with sufficient information to support facial expression recognition by simulation. In an additional orofacial motor examination that included attempted imitation of the six basic facial expressions, IMS 8 and 10 could execute a few (very) small facial movements. IMS eight was able to execute a mild combined backward/upward movement of the right angle of the mouth, a slight backward movement of the left angle of the mouth, a slight contraction of the mentalis and some slight movements of the upper eye lids. IMS10 was only able to execute a slight bilateral movement of the angles of the mouth backwards and downwards and some up and down movements of the superior eye lids. Importantly, however, these movements covered only a small part of the facial movements that would be required to imitate the different facial expressions of emotions (*Supplementary file 2*) and they were largely the same when they attempted to imitate the different facial expressions (sadness, happiness, anger, surprise, disgust, fear, see *Figure 2—figure supplement 2*). Control participants were unable to recognize video-clips showing the two IMS attempting to imitate the six same facial expressions (see *Figure 2—figure supplement 3*). In sum, the facial movements that the IMS were able to execute are small in amplitude, very limited in types, and do not discriminate different facial expressions.

Sixth, we investigated whether the discrepant results between IMS 8 and 10 and the other IMS, who did not meet the criteria for 'normotypical' performance (i.e., they performed below 0.85 standard deviation from the controls' mean performance in at least one experiment) could be because IMS 8 and 10 were affected by milder facial paralysis. This was clearly not the case: IMS 10 was among the participants with the most severe facial paralysis (see *Table 1*).

Instructively, and by contrast, the performance of the IMS who performed more poorly than controls in Experiments 1–3 was strongly and significantly correlated with their respective performance on the mid-level perception screening test (three *r*s > 0.5, *p*s < 0.05), and 7/9 of the IMS participants (IMS1, 2, 4, 5, 6, 7, 9) who failed to meet the criteria for normotypical efficient facial expression recognition obtained also equally weak or weaker performance in the facial identity and/or the emotional speech recognition task (see *Figure 1*). Thus, the IMS's performance variability seems to be at least partly due to associated visual and/or cognitive disorders, rather than to their facial paralysis. To further explore this possibility, we first performed a series of one-way ANCOVAs with Group as a fixed effect and participants' performance in Experiments 6–8 as covariates. Participants performing below two absolute deviations from the median of their group were discarded in order to satisfy ANCOVA normality assumptions (*Leys et al., 2013*). In line with the possibility that the IMS's performance variability is likely due to associated visual and/or cognitive disorders rather than to their facial paralysis, these analyses failed to reveal any significant effect of Group in Experiments 1, 2, 4 and 5 (all Fs <1; all *p*s > 0.37). The same analysis could not be performed on the results of Experiment three due to the low number of IMS who took part to this experiment. Then, to test more

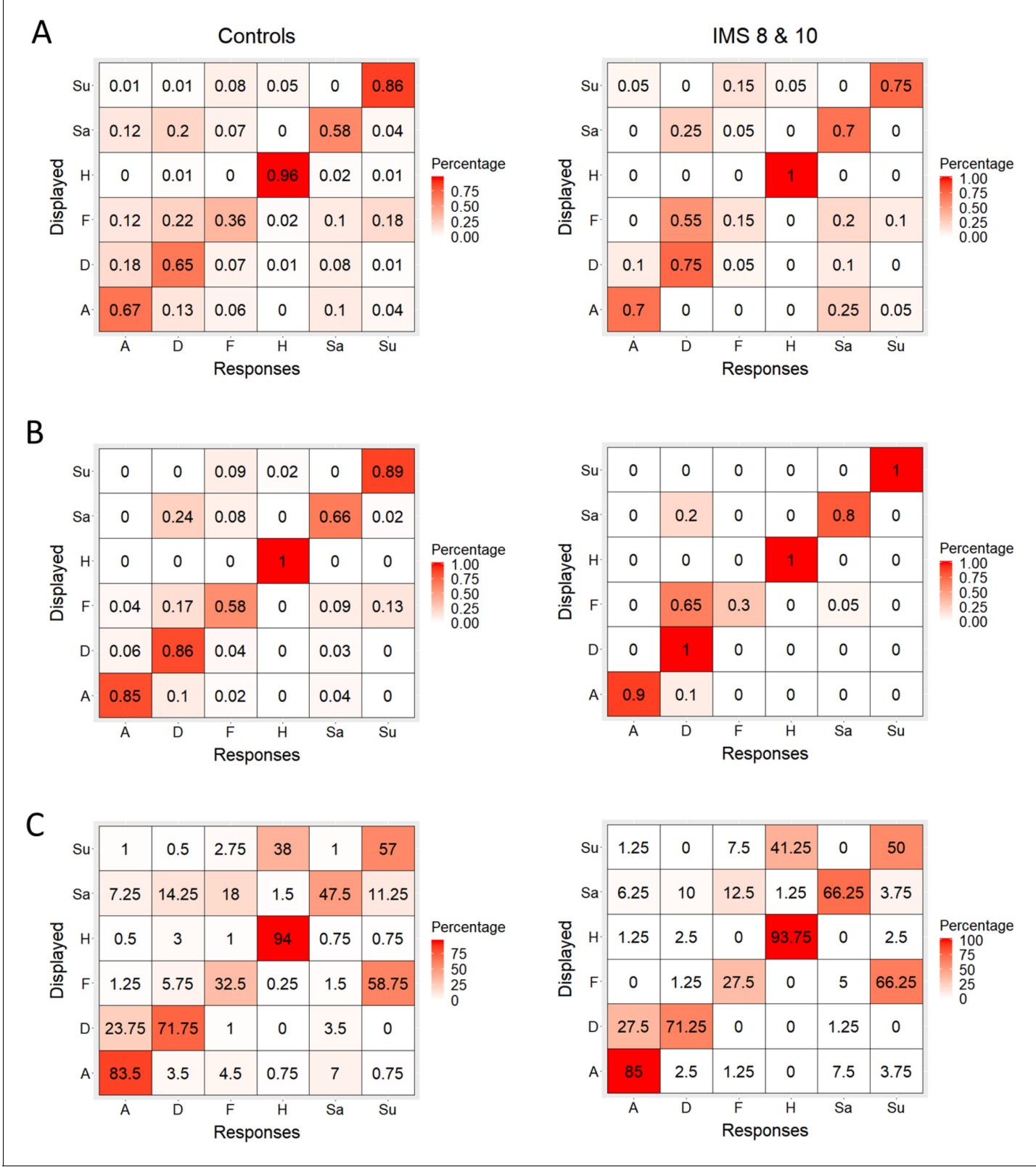

**Figure 2.** Confusion matrices. (A, B, C) Distribution of control participants' (left) and IMS 8 and 10's (right) percentage of trials in which they chose each of the six response alternatives when faced with the six displayed facial expressions in Experiment 1 (A), 2 (B), and 3 (C).
The online version of this article includes the following figure supplement(s) for figure 2:

*Figure 2 continued on next page*

*Figure 2 continued*

**Figure supplement 1.** Confusion matrices.
**Figure supplement 2.** Analysis of participants' action units when imitating facial expressions.
**Figure supplement 3.** IMS 8 and 10's facial expression execution.

directly the possibility that at least some IMS may have some facial expression recognition impairment that cannot be explained by a visual or cognitive deficit, we used *Crawford and Garthwaite, 2007* Bayesian Standardized Difference Test (BSDT) to test whether the performance of IMS1-7, 9 and 10 in at least some facial expression experiments may be comparatively less good than in the three control tasks. Only IMS1 performed significantly less well in Experiment five than in the three control experiments (Experiments 6–8). All other comparisons indicated either better performance in facial expression recognition experiments than in at least one of the control experiments (55% of the comparisons) or slightly but non-significantly weaker performance (45% of comparisons, all BSDTs > 0.1).

## Discussion

According to popular motor theories of facial expression recognition, efficient recognition of others' facial expressions cannot be achieved by visual analysis of the movements alone but requires additional unconscious covert imitation – motor simulation – of the observed movements. In this study, we tested a prediction drawn from these theories, namely, that any individual who has no motor representations of facial expressions due to congenital facial paralysis should be less efficient than typically developed individuals in recognizing facial expressions. We did this by assessing the performance of eleven individuals born with Moebius Syndrome, an extremely rare congenital non-progressive condition resulting in severe or complete bilateral facial paralysis often accompanied by visual and mid-level perceptual deficits, in five challenging facial expression experiments. We compared their performances to that of 25 young and highly educated control participants. As in previous studies (*Bate et al., 2013*; *Calder et al., 2000*; *Giannini et al., 1984*; *Nicolini et al., 2019*), several IMS failed to recognize facial expressions as efficiently as control participants in one (IMS 1, 3, 5, 7) or several (IMS 2, 6, 9, 11) experiments (see *Figure 1*). Although these difficulties have to be interpreted with caution because the performance of these IMS was compared to that of highly educated young adults who performed Experiments 3 and 4 at a higher level of performance than previously published norms, this finding suggests that some individuals with Moebius Syndrome have difficulty in recognizing facial expressions in at least some tasks. Such association of deficits is interesting but is difficult to interpret because of the co-occurrence of motor, visual and perceptual deficits in the IMS we tested (*Table 2*) makes it difficult to establish unambiguously the (possibly multiple) origins of these difficulties. Nevertheless, and more interestingly, we also found that two of

**Table 1.** Summary of the IMS participants' facial movements.

| IMS | Inf. lip | Sup. lip | Nose | Eyebrows | Forehead | R. cheek | L. cheek | Sup. R. eyelid | Inf. R. eyelid | Sup. L. eyelid | Inf. L. eyelid |
|---|---|---|---|---|---|---|---|---|---|---|---|
| IMS1 | None | None | None | None | None | None | None | None | None | None | None |
| IMS2 | Slight | None | None | None | None | Slight | Slight | Slight | Slight | Slight | Slight |
| IMS3 | Slight | None | None | None | None | None | None | Slight | None | Slight | None |
| IMS4 | Slight | None | None | None | None | Slight | Slight | None | None | None | None |
| IMS5 | Slight | Slight | None | None | None | Slight | Slight | Slight | None | Slight | None |
| IMS6 | Mild | None | None | None | None | Mild | Mild | Slight | None | Slight | None |
| IMS7 | None | None | None | None | None | Mild | None | Mild | Slight | None | None |
| IMS8 | Slight | None | None | None | None | Mild | Slight | Slight | Slight | Slight | Slight |
| IMS9 | Mild | None | None | None | None | Mild | Slight | Slight | Slight | None | None |
| IMS10 | None | None | None | None | None | Slight | Slight | None | None | None | None |
| IMS11 | None | None | None | None | None | Slight | None | None | None | None | None |

**Table 2.** Information regarding IMS participants' visual and visuo-perceptual abilities.

| | Vision | Reported best corrected acuity | Strabismus | Eye movements | Mid-level perception* (modified $t$-test)[2] |
|---|---|---|---|---|---|
| IMS1 | Hypermetropy, astigmatism | Mild vision loss (7/10) | Slight | H: Absent; V: Reduced | 0.9 |
| IMS2 | Hypermetropy, astigmatism | Normal vision | Slight | H: Absent; V: Reduced | −3.3 |
| IMS3 | Myopia | Normal vision | None | H: Absent; V: Reduced | −0.3 |
| IMS4 | Hypermetropy, astigmatism | Mild vision loss (5/10) | None | H: Absent; V: Typical | −2.9 |
| IMS5 | Hypermetropy, astigmatism | Mild vision loss (8/10) | Slight | H: Very limited; V: Typical | −0.3 |
| IMS6 | Hypermetropy, astigmatism | Normal vision | Slight | H: Typical; V: Typical | 0.1 |
| IMS7 | Hypermetropy, astigmatism | Moderate vision loss of the left eye (2/10) | None | H: Typical; V: Typical | 0.5 |
| IMS8 | Normal | Normal vision | None | H: Typical; V: Typical | 0.5 |
| IMS9 | Normal | Normal vision | Slight | H: Absent; V: Absent | −3.3 |
| IMS10 | Myopia | Mild vision loss (8/10) | None | H: Absent; V: Typical | −0.3 |
| IMS11 | Myopia, astigmatism | Mild vision loss: 6/10 right eye; 5/10 left eye | Slight | H: Reduced; V: Typical | −4.2 |

* Leuven Perceptual Organization Screening Test, L-POST (**Torfs et al., 2014**). [2](**Crawford and Howell, 1998**).

the IMS (IMS 8 and 10) fulfilled the criteria for 'normotypical efficient facial expression recognition': (1) they both scored above 0.85 standard deviations below the mean of the controls in all five facial expression recognition experiments; (2) they performed as well in these tasks as in other face-related visual tasks (face identity) and emotion recognition tasks (vocal emotion recognition); and (3) they performed these tasks in a way that is qualitatively similar to the controls. These two IMS had severe facial paralysis. They could not execute most of the facial movements that would be required to imitate the different facial expressions of emotions (**Supplementary file 2**). In addition, they executed largely the same inappropriate movements when they attempted to imitate the different facial expressions (sadness, happiness, anger, surprise, disgust, fear, see **Figure 2—figure supplements 2–3**). These dissociations challenge the central premise of the models of facial expression recognition centered on facial mimicry: they constitute existence proof that it is *possible* to account for efficient facial expression recognition without having to invoke a mechanism of 'motor simulation', even in very sensitive and challenging tasks requiring, for instance, the identification of a facial expression in 150 milliseconds, only from information available in the eye region, when the stimuli are morphs composed of a mixture of emotions, and when the task involves discriminating fake versus genuine smiles.

This conclusion is in line with previous reports of MS participants who achieved normal performance in facial expression recognition despite their congenital facial paralysis (**Bate et al., 2013**; **Calder et al., 2000**; **Rives Bogart and Matsumoto, 2010**). However, the interpretation of those findings was limited by the putative relative insensitivity of the facial expression recognition tasks used in those studies (**Van Rysewyk, 2011**; **De Stefani et al., 2019**). The findings reported herein go beyond this previous evidence and demonstrate that motor simulation contributes neither to the ease of facial expression recognition nor to its robustness in particularly challenging tasks.

Our findings are also in line with the results of previous behavioral studies showing that individuals congenitally deprived of hand motor representations nonetheless perceive and comprehend hand actions, which they cannot covertly imitate, as accurately, as fast, with the same biases, and very similar brain networks as typically developed participants (**Vannuscorps and Caramazza, 2015**; **Vannuscorps and Caramazza, 2016b**; **Vannuscorps and Caramazza, 2016c**; **Vannuscorps et al., 2019**; **Vannuscorps et al., 2012**). As such, beyond contributing to the question of the nature of the mechanisms involved in facial expression recognition, our findings are also relevant for theories of the types of representations that support action perception and recognition in general. According to motor simulation theories of action recognition, the recognition of others' actions cannot be achieved by visual analysis of the movements alone but requires unconscious covert imitation – motor simulation – of the observed movements (**Rizzolatti and Sinigaglia, 2010**). Our results challenge this premise.

In sum, our finding constitutes existence proof that the visuo-perceptual system can support efficient facial expression recognition unaided by motor simulation. Of course, this does not *imply* that motor simulation does not support facial expression in typically developed participants. Correlated sensorimotor experience may be necessary for the development of motor contributions to action perception (e.g., *Catmur and Heyes, 2019*) and IMS 8 and 10 may have developed an atypically efficient visual system to compensate for their congenital paralysis, for instance. However, the observation that these IMS performed the facial recognition tasks in a qualitatively similar manner to the controls makes this unlikely. In any event, and more importantly, in the current absence of compelling direct evidence that motor simulation does underlie facial expression recognition in typically developed participants, our results at the very least emphasize the need for a shift in the burden of proof regarding this question. If there were to be unambiguous evidence that efficient visual perception and interpretation of facial expressions requires the involvement of one's motor system this would revolutionize our understanding of how the Mind/Brain is organized and our results would have to be re-interpreted as useful evidence about the range of computational neural plasticity that is possible in a system that typically relies on motor simulation; but so far this claim is neither justified by current experimental evidence nor necessary to account for typically efficient facial expression recognition. At this juncture, facial expression recognition is thus better explained by the perceptual-cognitive view according to which it is supported by visuo-perceptual, structural and conceptual, but not sensorimotor, representations.

## Materials and methods

### Participants

We tested eleven individuals with severe to complete facial paralysis in the context of the Moebius Syndrome (IMS participants; eight females; all right-handed; various education levels; mean age ± SD: 27.7 ± 9.25 years) and compared their performance to that of 25 typically developed highly educated young adults (15 females; three left-handed; all college students or graduates without any history of psychiatric or neurological disorder; Mean age ± SD: 28.6 ± 6.5 years). All IMS were able to speak intelligibly and to convey emotions in their speech. Information regarding all participants' neurological and psychiatric history and about the IMS's medical, surgical and therapeutic history associated with the syndrome was obtained through a series of questionnaires (*Supplementary file 1*). Information about all participants' visual and mid-level perceptual skills were obtained through questionnaires and a perceptual screening test (*Table 2*). Information about the IMS's facial motor abilities was obtained through a facial movements examination (see supplemental methods for detail; *Table 1*).

### Materials and procedures

The experimental investigations were carried out from October 2015 to April 2019 in sessions lasting between 60 and 90 min. The study was approved by the biomedical ethics committee of the Cliniques Universitaires Saint-Luc, Brussels, Belgium and all participants gave written informed consent prior to the study.

Participants performed five facial expression recognition tasks, two facial identity recognition tasks, and an emotional speech recognition task. These experiments were controlled by the online testable.org interface (http://www.testable.org), which allows precise spatiotemporal control of online experiments. Control participants were tested on the 15.6-inch anti-glare screen (set at 1366 × 768 pixels and 60 Hz) of a Dell Latitude E5530 laptop operated by Windows 10. During the experiment, they sat comfortably at a distance of about 60 cm of the screen. The IMS were tested remotely under supervision of the experimenter through a Visio conference system (Skype). They sat at about 60 cm from their own computer screen. All procedures were the same for all participants. At the beginning of each experiment, the participant was instructed to set the browsing window of the computer to full screen, minimize possible distractions (e.g., TV, phone, etc.) and position themselves at arm's length from the monitor for the duration of the experiment. Next, a calibration procedure ascertaining homogeneous presentation size and time on all computer screens took place.

Experiments 1 and 2 tested whether motor simulation supports efficient recognition of briefly-presented facial expressions. In each of the 60 trials of Experiments 1 and 2, participants viewed a

picture of an actor's face expressing one of six facial expressions (anger, disgust, fear, happiness, sadness, and surprise; *Lundqvist et al., 1998*) presented for either 150 ms (Experiment 1) or 1 s (Experiment 2) between two pictures of the same actor displaying a neutral expression (presented for 1 s each). Participants were asked to carefully observe the target picture and to associate it with its corresponding label, presented among five alternatives.

Experiments 3 and 4 tested whether motor simulation enhances stimulus identification under adverse perceptual conditions, such as when the emotion must be inferred from minimal information or when it is subtle and ambiguous. In Experiment 3, participants viewed 96 video clips depicting an actor's face expressing one of the six basic emotions at one of four different levels of intensity (40%, 60%, 80%, and 100%; *Kessels et al., 2014*; *Montagne et al., 2007*) and had to associate it with its corresponding label among six alternatives. This experimental paradigm has been shown to be sensitive to subtle emotion recognition disorders in clinical conditions such as post-traumatic stress disorder (*Poljac et al., 2011*), amygdalectomy (*Ammerlaan et al., 2008*) and social phobia (*Montagne et al., 2006*).

In Experiment 4, we tested the ability to recognize more complex mental states (e.g., skeptical, insisting, suspicious) from information restricted to the actor's eye region. On each of the 36 trials of this experiment (the 'Reading the Mind in the Eyes' test Revised version; *Baron-Cohen et al., 2001*), participants were asked to associate a picture depicting the eye-region of someone's face to the verbal label that best described that person's emotion or mental state (e.g., skeptical, insisting, suspicious) among four subtly different alternatives. Performance in this experiment has been previously shown to be sensitive to both subtle experimental modulations of the observers' motor system and slight emotion recognition disorders in clinical conditions: It is affected by botulinum toxin (Botox) injections on the forehead (*Neal and Chartrand, 2011*), oxytocin administration (*Domes et al., 2007*), and hampered in persons with autism (*Baron-Cohen et al., 2001*), schizophrenia (*Kettle et al., 2008*), anorexia nervosa (*Harrison et al., 2010*), depression (*Szanto et al., 2012*), Parkinson's disease (*Tsuruya et al., 2011*) and Huntington's disease (*Allain et al., 2011*).

Experiment five tested the hypothesis that motor simulation is particularly important for the ability to discriminate fake versus genuine smiles (e.g., *Paracampo et al., 2017*). In each of the 32 trials of this experiment, participants viewed a video clip showing an actor who was either spontaneously smiling out of amusement (genuine smile) or producing a forced (fake) smile and had to use subtle morphological and dynamic features of the smile in order to discriminate genuine from fake smiles (*Ekman, 1982*; *Krumhuber and Manstead, 2009*; *Paracampo et al., 2017*). Performance in the Amusement from Smiles task is influenced by simple procedures such as wearing a mouth guard (*Rychlowska et al., 2014*), TMS in the face sensorimotor cortex (*Paracampo et al., 2017*), and vowel articulation (*Ipser and Cook, 2016*).

Experiments 6 and 7 tested participants' facial identity recognition skills. Experiment six used the Cambridge Face Memory Test (CFMT; *Duchaine and Nakayama, 2006*), which has two parts. In the first phase, participants were asked to observe, memorize, and recognize six persons' faces. For each person, they were presented with three pictures of the person's face displayed from three different viewpoints for 3 s each and they were then asked to recognize this face among two foils in three successive forced-choice trials. In the second phase, participants were first presented with a frontal view of these six faces for 20 s, and then asked to recognize any of these six faces from a set of 30 three-alternative forced-choice trials containing new images of faces studied in the first phase. Performance was measured by counting the number of trials, out of 48, in which the learned face was chosen correctly. In each of the 8 trials of Experiment 7 – the Cambridge Face Perception Test (CFPT; *Duchaine et al., 2007*) – participants were presented with a target face and, below it, six pictures morphed to contain different proportions of the target face (28%, 40%, 52%, 64%, 76% and 88%) and were asked to sort them from most like to least like the target face. For each trial, the final order was scored by summing the deviations from the correct order (e.g., if a face is three places away from its correct place, it is given a score of 3).

Experiment eight tested participants' ability to recognize emotions from emotional speech. In each of the 60 trials of this experiment, participants were asked to listen to a ~ 2 s long audio-clip in which a female or a male speaker simulated one of seven emotions (anger, anxiety/fear, boredom, disgust, happiness, sadness, neutral) while producing a sentence which could be applied to any emotion in a language that none of the IMS or control participants declared to understand (in German; from *Burkhardt et al., 2005*). Participants were asked to listen carefully to the audio-clips and, then,

to pick its corresponding label presented among the six other alternatives. We counted each participant's number of correct responses.

The participants with Moebius Syndrome additionally participated to a motor, which included three main parts: (1) They were asked to imitate facial movements involving specific muscles on the territory of the facial nerve: the buccinator (pressing cheeks on the side of the teeth and pull back the angle of the mouth), zigomaticus major (large smile), risorius (draw the angle of the mouth straight backwards), levator anguli oris (pulling the angle of the mouth upward), depressor anguli oris (pulling the angle of the mouth downward), levator labii superioris (raise and protrude the upper lips), depressor labii inferioris (pull the lower lip and angle of the mouth down), the mentalis (protrude the lower lip and make a pouting face), the frontalis (elevate the eyebrows), the corrigator supercilia and depressor supercilia (frown), orbicularis oculi (close the eyes gently, close the eyes tightly), and the procerus and nasalis (wrinkle the nose) muscles. (2) In order to test the trigeminal and hypoglossal nerves, we asked them to produce various facial other movements upon verbal command: opening and closing the jaws, moving the tongue from left to right, up and down, back and forth, and articulating repeatedly the syllable/pa/for 5–6 s. (3) In order to document their ability to execute facial expressions, they were asked to imitate as well as possible six facial expressions (happiness, anger, fear, disgust, sadness and surprise) based on a picture example from the Karolinska Directed Emotional Faces set (Actress AF01, front view; Lundqvist, D., Flykt, A., and Öhman, A., 1998). Their performance during these tasks was recorded and analyzed offline. The results of these analyses are presented on *Table 1*.

## Acknowledgements

This research was supported by the Mind, Brain and Behavior Interfaculty Initiative provostial funds. MA is a research associate at the Fonds National de la Recherche Scientifique (FRS-FNRS, Belgium).

## Additional information

### Funding

| Funder | Grant reference number | Author |
| --- | --- | --- |
| Harvard University | Mind, Brain and Behavior Interfaculty Initiative | Alfonso Caramazza |
| Institut national de la recherche scientifique | | Michael Andres |

The funders had no role in study design, data collection and interpretation, or the decision to submit the work for publication.

### Author contributions

Gilles Vannuscorps, Conceptualization, Formal analysis, Investigation, Methodology, Writing - original draft, Project administration; Michael Andres, Alfonso Caramazza, Conceptualization, Supervision

### Author ORCIDs

Gilles Vannuscorps (iD) https://orcid.org/0000-0001-5686-7349

### Ethics

Human subjects: The study was approved by the local Ethical committee at UCLouvain (Registration # B403201629166). Written informed consents were obtained from all participants prior to the study, and after the nature and possible consequences of the studies were explained.

### Decision letter and Author response

Decision letter https://doi.org/10.7554/eLife.54687.sa1
Author response https://doi.org/10.7554/eLife.54687.sa2

# Additional files

## Supplementary files

• Supplementary file 1. Information regarding IMS participants' demographic, neurological, psychiatric, medical and surgical/therapeutic history.

• Supplementary file 2. Facial action units corresponding to the facial expression of the six basic emotions and their presence/absence in the repertoire of the IMS 8 and 10.

• Transparent reporting form

## Data availability

Data and stimulus materials are publicly available and can be accessed on the Open Science Framework platform (https://doi.org/10.17605/OSF.IO/8T4FV).

The following dataset was generated:

| Author(s) | Year | Dataset title | Dataset URL | Database and Identifier |
|---|---|---|---|---|
| Vannuscorps G, Andres M, Caramazza A | 2020 | Data from: Efficient recognition of facial expressions does not require motor simulation | https://doi.org/10.17605/OSF.IO/8T4FV | Open Science Framework, 10.17605/OSF.IO/8T4FV |

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
