## [Decision Letter]

**Acceptance summary:**

An influential theory of face perception builds on the general concept of embodied cognition, with the idea that this perceptual process requires the engagement of more motoric brain regions, those involved in the generation of facial expressions. By this view, recognition might entail the operation of simulation, where the perceiver simulates the action that might best match the sensory input to identify the affective intent. As a test of this hypothesis, the current study turns to an atypical population, individuals with Moebius Syndrome, a condition marked by congenital facial paralysis. The results show that some of these individuals exhibit normal performance on facial expression recognition, despite an absence of facial motor representations. As such, the results present an intriguing counter-example to the claim that facial expression recognition requires referent to motoric representations.

**Decision letter after peer review:**

Thank you for submitting your article "Efficient recognition of facial expressions does not require motor simulation" for consideration by *eLife*. Your article has been reviewed by three peer reviewers, and the evaluation has been overseen Richard Ivry as the Senior Editor and Reviewing Editor. The following individual involved in review of your submission has agreed to reveal their identity: Adrienne Wood.

The reviewers have discussed the reviews with one another and the Reviewing Editor has drafted this decision to help you prepare a revised submission. As you will see, there are no serious issues for revision. It is mostly about the theoretical framing and the weight to be given individual cases-issues we have covered to some extent in our initial email correspondence. They did come up again in review and I've consolidated things so you have the feedback from some new "samples." But I don't expect they will require much revision on your part.

Summary:

The study aims to assess whether individuals who suffer from Moebius syndrome (IMS) can achieve normotypical efficient facial expression recognition despite the congenital absence of relevant facial motor representations (a necessity hypothesis). The manuscript is well written. The research design is well thought out and uses a very high standard for the assessment of normotypical performance. The results show that some of the IMS patients perform well within the normal range on the emotional facial recognition tests even though they are unable to produce such expressions. As such, they present examples arguing against the necessity hypothesis, challenging simulation-based models of facial recognition. The study obviously suffers from a small sample, but this is a rare population.

Major Comments for Revision:

1) Theoretical framing:

a) Concern that the presentation of the motor theories of action perception are a straw man. Question the strong claim that the "standard belief" in the field is that sensorimotor activity is necessary for action perception.

b) More recent accounts of mimicry and the closely-related idea of sensorimotor simulation emphasize how motor-perceptual mappings could emerge through associative learning. If, over the course of development, an infant is regularly experiencing the correlation between their own facial movements and the visual perception of others' facial expressions (because adults are engaging in extensive mimicry), then they could learn to recruit this cross-modal association in the service of efficient interpretation of others' facial expressions. But if the infant never experiences that correlation-for instance, if their facial muscles are impaired-then that association won't be made and won't be drawn on to process the meaning of others' facial expressions. We agree that it is unlikely that perception of actions would require motor activity; humans can learn to recognize the differences between thousands of bird species without sensorimotor simulation, so if necessary, they could rely on purely visual activity to recognize the differences between others' facial movements. A fairer Introduction and Discussion would acknowledge these more nuanced sensorimotor perspectives, which are compatible with the present findings (see de Klerk et al., 2018; Catmur and Heyes, 2019; Koban et al., 2019).

c) The "necessity hypothesis" is ascribed to articles that do not make such claims (Wood et al., 2016; Goldman and Sripada, 2005). Wood et al., 2016, argue simply that sensorimotor activity can support facial expression processing, saying, "whether chronic facial paralysis disrupts emotion recognition or compensatory perceptual strategies can eventually develop is not fully clear". Similarly, Goldman and Sripada, 2005 write about Moebius patients, "…given the long-standing nature of their impairments, these subjects' normal performance may reflect the operation of a compensatory strategy." So at least these two papers are not properly represented in the present manuscript.

2) Weight to be given to individual cases, especially when they are the exception, not the rule:

Does the presence of two individuals with Moebius Syndrome performing within normal bounds contradict contemporary sensorimotor perspectives? And even here, the pattern is more of a trend. As shown in Figure 1, patients 8 and 10 are the top two IMS performers in three of the five expression recognition tasks.

3) Selectivity of impairment:

a) The IMS group perform less well overall than the controls in the facial emotion recognition. They also tend to perform less well on other tasks as well including emotion recognition from speech.

b) There is a high bar here for rejecting the necessity hypothesis, using a larger alpha (any p <.20 would be taken as evidence that the Moebius patients are different from the healthy controls), discarding the low-performing healthy controls, and including the non-facial expression tasks. Moebius syndrome can have other symptoms besides facial motor impairment-importantly, visual impairment-that might account for deficits in perception tasks. Indeed, there is a correlation between the patients' mid-level visual screening test performance and their performance on the facial expression tasks, suggesting any group differences might be due to visual, not motor, disruptions.

4) Additional analyses:

Consider a between-group comparison of performance on the expression recognition tasks, since that's the analysis most people would expect. This could even include performance on the facial identity tasks as a covariate to control for "general" performance on such tasks.

[Editors' note: further revisions were suggested prior to acceptance, as described below.]

Thank you for submitting your article "Efficient recognition of facial expressions does not require motor simulation" for consideration by *eLife*. I felt it was only important to engage one of the original reviewers, Adrienne Wood, to assist me in evaluating the revision.

We think the paper is essentially there but Adrienne has one substantive request and I find it reasonable. To quote her:

I appreciate the authors' thorough responses to our comments on their manuscript. I continue to think that in an effort to carefully define and then reject the motor simulation hypothesis, they have shifted the burden of proof too far in the other direction. As we agreed originally, I do think the case study nature of their approach is ok. If they can find any IMS with normal expression recognition, then it brings into question the strong motor simulation position that motor functioning is “necessary” for expression recognition on challenging tasks. But I think they go too far when they reject even running the between-group statistical test (reviewer point #4) that might create a more nuanced picture. Perhaps IMS often suffer expression recognition impairment (even controlling for more general visual impairment), but not always. The authors write, "we would prefer to limit the statistical analysis presented in the paper to only those meaningful for the question at hand: is it possible to achieve efficient facial expression recognition without motor simulation." There is no rule that secondary questions cannot be answered in a Results section; they could identify those analyses as "post-hoc" if they are concerned about fishing.

---

## [Author Response]

Major Comments for Revision:1) Theoretical framing:a) Concern that the presentation of the motor theories of action perception are a straw man. Question the strong claim that the "standard belief" in the field is that sensorimotor activity is necessary for action perception.

The prediction that we tested is not simply whether motor simulation is “necessary for action perception”. The experiments reported in this study were specifically designed to test the more fine-grained hypothesis that motor simulation is necessary for *efficient* (i.e., fast and accurate) facial expression recognition, in particular when the tasks are challenging, such as when facial expressions must be interpreted after only a short exposure (Experiment 1 and 2), when they are more complex or ambiguous (Experiment 3 and 4) or when the task requires subtle intra-category discriminations such as discriminating fake versus genuine smiles, like it is predicted by numerous authors (see below). There is no doubt that this hypothesis is currently widespread in the field. To quote only some of the most recent and influential articles, for instance:

Niedenthal et al., 2010, propose that “Embodied simulation supports the recognition and access to meaning of facial expressions”. They further clarify their position and explain that although “recognition tasks on prototypic expressions can be accomplished by perceptual analysis alone, without motor mimicry (….) in some cases, however, mimicry does facilitate recognition”. They go on providing three examples of tasks in which motor simulation may contribute to facial expression recognition: “Niedenthal et al., 2001, for example, observed effects of mimicry when participants had to detect the boundary of facial expression between happiness and sadness. In a more recent study, Stel and van Knippenberg, 2008 found that blocking mimicry affected the speed, but not the accuracy, of categorizing facial expressions as positive or negative. These ﬁndings point to the possibility that simulation does become important in recognition tasks when they require ﬁne distinctions in smile meaning, such as the processing of different smile types.” These three hypotheses were specifically tested in our Experiments 3, 1 and 2, and 5, respectively.

Wood et al., 2016’s paper “Sensorimotor Simulation Contributes to Facial Expression Recognition”, proposes that “When we observe a facial expression of emotion, we often mimic it. This automatic mimicry reflects underlying sensorimotor simulation that supports accurate emotion recognition.” In the paragraph titled “Simulation and the Recognition of Facial Expression”, they note that “Substantial research evidence suggests that sensorimotor simulation contributes to accurate and efficient recognition of the specific emotion, valence, intensity and intentionality conveyed by facial expressions.” And, they go on with “Factors that disrupt somatosensory feedback from, or motor output to, the facial muscles are expected to affect emotion recognition.” and continue noting that motor simulation may be particularly useful precisely for the types of tasks that we presented to our participants: “Sensorimotor simulation may be especially useful for emotion recognition when the perceived expression is subtle or ambiguous, or the judgment to be made particularly challenging. Judging the authenticity of a smile, for instance, is a complex task that relies on detecting subtle differences in the morphological specificities and temporal dynamics of the smile. (…). Disrupting mimicry impairs the ability of observers to distinguish between videos of people producing spontaneous and intentional smiles” They end the paragraph with “to summarize the current evidence, automatic sensorimotor simulation plays a functional role not only in the recognition of actions, but also in the processing of emotional expressions”.

Paracampo et al., 2016’s paper “sensorimotor network crucial for inferring amusement from smiles” argues that “ motor and somatosensory circuits for controlling and sensing facial movements are causally essential for inferring amusement from another’s smile”, once again a prediction that applies precisely to the ability that we tested in one of our experiments (Experiment 5). Note that, in fact, we used precisely the same stimuli as Paracampo and colleagues.

Ipser and Cook, 2016, conclude their article by emphasizing that their findings “are consistent with models proposing that the motor system makes *a causal contribution* to the perception and interpretation of facial expressions”.

We think the paper is clear about the hypotheses that we tested. When presenting the motor simulation theories of facial expression recognition, for instance, we wrote that they assume that “*efficient* (i.e., fast and accurate) facial expression recognition cannot be achieved by visual analysis *alone* but requires a process of motor simulation”. At a later point we are even more specific and wrote “previous findings left open the possibility that motor simulation may contribute to facial expression recognition efficiency when the tasks are more challenging, such as when facial expressions are more complex, must be interpreted quickly, only partial information is available, or when the task requires subtle intra-category discriminations such as discriminating fake versus genuine smiles. The research reported here was designed to explore this large remaining hypothesis space for a role of motor simulation in facial expression recognition: is it possible to achieve efficient facial expression recognition without motor simulation in the type of sensitive and challenging tasks that have been cited by proponents of the motor theories as examples of tasks in which motor simulation is needed to support facial expression recognition?”.

b) More recent accounts of mimicry and the closely-related idea of sensorimotor simulation emphasize how motor-perceptual mappings could emerge through associative learning. If, over the course of development, an infant is regularly experiencing the correlation between their own facial movements and the visual perception of others' facial expressions (because adults are engaging in extensive mimicry), then they could learn to recruit this cross-modal association in the service of efficient interpretation of others' facial expressions. But if the infant never experiences that correlation-for instance, if their facial muscles are impaired-then that association won't be made and won't be drawn on to process the meaning of others' facial expressions. We agree that it is unlikely that perception of actions would require motor activity; humans can learn to recognize the differences between thousands of bird species without sensorimotor simulation, so if necessary, they could rely on purely visual activity to recognize the differences between others' facial movements. A fairer Introduction and Discussion would acknowledge these more nuanced sensorimotor perspectives, which are compatible with the present findings (see de Klerk et al., 2018; Catmur and Heyes, 2019; Koban et al., 2019).

In this paper, we tested whether congenital facial paralysis hampers facial expression recognition efficiency in challenging tasks, as is predicted by various theories of facial expression recognition based on motor simulation (e.g., the generate-and-test model, the reverse simulation model, Niedenthal and coworkers’ simulation of smiles hypothesis). The reviewer asks us to acknowledge the possibility that motor simulation could be necessary for efficient facial expression recognition (would be “drawn on to process the meaning of others' facial expressions”) *only* in people in whom associative learning has formed connections between sensory and motor representations of actions.

We followed this suggestion, we now write “In sum, our finding constitutes existence proof that the visuo-perceptual system can support efficient facial expression recognition unaided by motor simulation. Of course, this does not imply that motor simulation does not support facial expression in typically developed participants. For example, correlated sensorimotor experience may be necessary for the development of motor contributions to action perception (e.g., Catmur and Heyes, 2019) and IMS 8 and 10 may have developed an atypically efficient visual system to compensate for their congenital paralysis.”

Importantly, however, there is currently no evidence that the IMS would perform the task differently and no empirical evidence that motor simulation is indeed necessary for individuals who have benefitted from correlated visuo-motor experience. Hence, our results at the very least emphasize the need for a shift in the burden of proof regarding this question.

c) The "necessity hypothesis" is ascribed to articles that do not make such claims (Wood et al., 2016; Goldman and Sripada, 2005). Wood et al., 2016, argue simply that sensorimotor activity can support facial expression processing, saying, "whether chronic facial paralysis disrupts emotion recognition or compensatory perceptual strategies can eventually develop is not fully clear". Similarly, Goldman and Sripada, 2005, write about Moebius patients, "…given the long-standing nature of their impairments, these subjects' normal performance may reflect the operation of a compensatory strategy." So at least these two papers are not properly represented in the present manuscript.

We cite these two papers in the Introduction : “an alternative view suggesting that efficient (i.e., fast and accurate) facial expression recognition cannot be achieved by visual analysis alone but requires a process of motor simulation – an unconscious, covert imitation of the observed facial postures or movements – has gained considerable prominence”.

We decided to cite Goldman and Sripada, 2005, on the ground that they discuss two potential simulationist models (the generate-and-test model on p202 and the reverse simulation model on p203, see Figure 2 on P. 204) and emphasize (p. 209) that these two models predict “reduced recognition” of facial expressions in case of a decrease of motor resources (e.g., dual task paradigm). The prediction that “reduced recognition” should result from a decrease of motor resources follows necessarily from the claim that motor simulation is necessary for efficient facial expression recognition.

Even though Wood et al., 2016, wrote “whether chronic facial paralysis disrupts emotion recognition or compensatory perceptual strategies can eventually develop is not fully clear”, we decided to cite that paper on the ground that, on the first page of the article, they write : “People’s recognition and understanding of others’ facial expressions is compromised by experimental (e.g., mechanical blocking) and clinical (e.g., facial paralysis and long-term pacifier use) disruption to sensorimotor processing in the face. Emotion perception involves automatic activation of pre and primary motor and somatosensory cortices and the inhibition of activity in sensorimotor networks reduces performance on subtle or challenging emotion recognition tasks. Sensorimotor simulation flexibly supports not only conceptual processing of facial expression but also, through cross-modal influences on visual processing, the building of a complete percept of the expression”. The interfering, automatic and constitutive nature of the role assigned to motor simulation by the authors in their article led us to assume that they view motor simulation as necessary for efficient recognition of facial expression.

Admittedly, both Goldman and Sripada, 2005, and Wood et al., 2016, mention that “compensatory strategies” may develop in individuals with facial paralysis. However, as this claim is not supported by any evidence (or explanation of what that compensatory strategy could be or why it would not develop in typically developed people) it seems to serve no other purpose than insulating the motor simulation hypothesis from the implications of the performance of these individuals. Hence, while the authors refer to “compensatory strategies” to discard incompatible evidence from patients with facial paralysis, they nevertheless cite evidence that “People’s recognition and understanding of others’ facial expressions is compromised by experimental (e.g., mechanical blocking) and clinical (e.g., facial paralysis and long-term pacifier use) disruption to sensorimotor processing in the face” as evidence in favor of the same theory.

In the revised version of the manuscript we nevertheless decided to follow the reviewer’s suggestion and delete the reference to Wood et al., 2016, and we took the liberty of replacing it with a more recent paper with a less ambiguous stand on the role of motor simulation in the recognition of motor actions (Paracampo et al., 2017).

2) Weight to be given to individual cases, especially when they are the exception, not the rule:Does the presence of two individuals with Moebius Syndrome performing within normal bounds contradict contemporary sensorimotor perspectives? And even here, the pattern is more of a trend. As shown in Figure 1, patients 8 and 10 are the top two IMS performers in three of the five expression recognition tasks.

We identified two separate issues in this comment.

The first concerns whether “the presence of two individuals with Moebius Syndrome performing within normal bounds contradict contemporary sensorimotor perspectives?”. It depends what the reviewer means by “contradict”. As written in the final paragraph of the Discussion, we propose that this finding constitutes existence proof that the visuo-perceptual system can support efficient facial expression recognition unaided by motor simulation. However, this observation does not necessarily *imply* that motor simulation does not support facial expression in typically developed participants. Nevertheless, in the absence of convincing evidence that motor simulation is used in the normotypical population, this finding emphasizes the importance of a shift in the burden of proof regarding the role of motor simulation for facial expression recognition.

The second concerns the weight to be given to individual cases vs the group (or the other individuals), especially when the cases are the exception, not the rule. We suggest that the issue should be considered on a case-by-case basis: one should ask which result constitutes the most valid constrain for the theories tested and what is the most plausible account of the discrepancy between the cases and the rest of the group.

Concluding that facial paralysis hampers facial expression recognition based on the result of the 9 IMS who perform among the weak participants (*p < 0.2)* in at least one experiment would require that (1) their weak performance can be related specifically to their facial paralysis and (2) that a plausible account of the performance of the two best IMS compatible with the simulation thesis can be formulated. In the paper, however, we found no empirical support for either (1) or (2). In contrast to (1), we report that the weak performance of the 9 IMS in facial expressions tasks is likely not due to their facial paralysis but rather, to their associated visual and/or cognitive disorder (Results final paragraph). In contrast to (2), we found that the performance of IMS 8 and 10 cannot be explained by them being extraordinarily efficient individuals across-the-board (In comparison to control participants, these two individuals were as efficient in facial expression recognition experiments than in other experiments, see Results paragraph three) or by them being less severely paralyzed (Results paragraph six).

In sum, in the specific case of our study, the data from the 9 IMS provides no support for the motor simulation theory. In contrast, the results of IMS8 and 10 provide unambiguous evidence that efficient facial expression recognition can be achieved in the absence of facial movements.

3) Selectivity of impairment:a) The IMS group perform less well overall than the controls in the facial emotion recognition. They also tend to perform less well on other tasks as well including emotion recognition from speech.

This is correct. As written: “Moebius syndrome typically impacts not only the individuals’ sensorimotor system, but also their visual, perceptual, cognitive, and social abilities.” Therefore, a weaker than normal average performance across-the-board is expected.

b) There is a high bar here for rejecting the necessity hypothesis, using a larger alpha (any p <.20 would be taken as evidence that the Moebius patients are different from the healthy controls), discarding the low-performing healthy controls, and including the non-facial expression tasks. Moebius syndrome can have other symptoms besides facial motor impairment-importantly, visual impairment-that might account for deficits in perception tasks. Indeed, there is a correlation between the patients' mid-level visual screening test performance and their performance on the facial expression tasks, suggesting any group differences might be due to visual, not motor, disruptions.

We thank the reviewer for this positive assessment of our approach. Setting such as high bar allowed us to minimize the likelihood of false negatives (i.e., the risk to conclude erroneously that an IMS achieves a “normotypical level” of efficiency) and provided more power to detect even subtle degradations of facial expression recognition skills.

4) Additional analyses:Consider a between-group comparison of performance on the expression recognition tasks, since that's the analysis most people would expect. This could even include performance on the facial identity tasks as a covariate to control for "general" performance on such tasks.

We thank the reviewer for this suggestion but we would prefer to limit the statistical analysis presented in the paper to only those meaningful for the question at hand: is it possible to achieve efficient facial expression recognition without motor simulation. We are already clear about this in the paper (Introduction section).

[Editors' note: further revisions were suggested prior to acceptance, as described below.]

We think the paper is essentially there but Adrienne has one substantive request and I find it reasonable. To quote her:I appreciate the authors' thorough responses to our comments on their manuscript. I continue to think that in an effort to carefully define and then reject the motor simulation hypothesis, they have shifted the burden of proof too far in the other direction. As we agreed originally, I do think the case study nature of their approach is ok. If they can find any IMS with normal expression recognition, then it brings into question the strong motor simulation position that motor functioning is “necessary” for expression recognition on challenging tasks. But I think they go too far when they reject even running the between-group statistical test (reviewer point #4) that might create a more nuanced picture. Perhaps IMS often suffer expression recognition impairment (even controlling for more general visual impairment), but not always. The authors write, "we would prefer to limit the statistical analysis presented in the paper to only those meaningful for the question at hand: is it possible to achieve efficient facial expression recognition without motor simulation." There is no rule that secondary questions cannot be answered in a Results section; they could identify those analyses as "post-hoc" if they are concerned about fishing.

We decided to follow the reviewer and the Editor’s suggestion to include between-group statistical tests in the result section of the manuscript devoted to discussing whether the IMS’s performance variability is most likely due to associated visual and/or cognitive disorders or to their facial paralysis. We have now added:

“To further explore this possibility, we first performed a series of one-way ANCOVAs with Group as a fixed effect and participants’ performance in Experiments 6-8 as covariates. Participants performing below 2 absolute deviations from the median of their group were discarded in order to satisfy ANCOVA normality assumptions (Leys, Ley, Klein, Bernard and Licata, 2013). In line with the possibility that the IMS’s performance variability is likely due to associated visual and/or cognitive disorders rather than to their facial paralysis, these analyses failed to reveal any significant effect of Group in Experiments 1, 2, 4 and 5 (all Fs < 1; all *p*s > 0.37). The same analysis could not be performed on the results of Experiment 3 due to the low number of IMS who took part to this experiment.”

In addition, in order to address more directly the possibility raised by the reviewer that “Perhaps IMS often suffer expression recognition impairment (even controlling for more general visual impairment), but not always”, we now report the results of a new set of single case analyses allowing to test this possibility:

“Then, to test more directly the possibility that at least some IMS may have some facial expression recognition impairment that cannot be explained by a visual or cognitive deficit, we used Crawford and Garthwaite’s, 2007, Bayesian Standardized Difference Test (BSDT) to test whether the performance of IMS1-7, 9 and 10 in at least some facial expression experiments may be comparatively less good than in the three control tasks. Only IMS1 performed significantly less well in Experiment 5 than in the three control experiments (Experiments 6-8). All other comparisons indicated either better performance in facial expression recognition experiments than in at least one of the control experiments (55% of the comparisons) or slightly but non-significantly weaker performance (45% of comparisons, all BSDTs > 0.1).”

In sum, these two additional series of analyses corroborate the conclusion that “the IMS’s performance variability seems to be at least partly due to associated visual and/or cognitive disorders, rather than to their facial paralysis”.